# Food Insecurity and Health Outcomes Other than Malnutrition in Southern Africa: A Descriptive Systematic Review

**DOI:** 10.3390/ijerph19095082

**Published:** 2022-04-21

**Authors:** Elias M. A. Militao, Elsa M. Salvador, Olalekan A. Uthman, Stig Vinberg, Gloria Macassa

**Affiliations:** 1Department of Health Sciences, Faculty of Humanities, Mid Sweden University, Holmgatan 10, 851 70 Sundsvall, Sweden; stig.vinberg@miun.se (S.V.); gloria.macassa@hig.se (G.M.); 2Department of Public Health and Sports Science, Faculty of Occupational and Health Sciences, University of Gävle, Kungsbacksvägen 47, 801 76 Gävle, Sweden; 3Department of Biological Sciences, Faculty of Science, Eduardo Mondlane University, 3453 Julius Nyerere Avenue, Maputo 257, Mozambique; elsamariasalvador@gmail.com; 4Warwick Centre for Global Health, Division of Health Sciences, Warwick Medical School, University of Warwick, Coventry CV4 7AL, UK; olalekan.uthman@warwick.ac.uk; 5Division of Epidemiology and Biostatistics, Department of Global Health, Faculty of Health Sciences, Stellenbosch University, Francie van Zijl Drive, Tygerberg, Cape Town 7505, South Africa; 6EPI Unit, Instituto de Saúde Pública, Universidade do Porto, Rua das Taipas 135, 4050-600 Porto, Portugal

**Keywords:** food insecurity, health outcomes, measurement, southern Africa

## Abstract

Food insecurity (FI) is one of the major causes of malnutrition and is associated with a range of negative health outcomes in low and middle-income countries. The burden of FI in southern Africa is unknown, although FI continues to be a major public health problem across sub-Saharan Africa as a whole. Therefore, this review sought to identify empirical studies that related FI to health outcomes among adults in southern Africa. Altogether, 14 publications using diverse measures of FI were reviewed. The majority of the studies measured FI using modified versions of the United States Department of Agriculture Household Food Security Survey Module. A wide range in prevalence and severity of FI was reported (18–91%), depending on the measurement tool and population under investigation. Furthermore, FI was mostly associated with hypertension, diabetes, anxiety, depression and increased risk of human immunodeficiency virus (HIV) acquisition. Based on the findings, future research is needed, especially in countries with as yet no empirical studies on the subject, to identify and standardize measures of FI suitable for the southern African context and to inform public health policies and appropriate interventions aiming to alleviate FI and potentially improve health outcomes in the region.

## 1. Introduction

Food insecurity (FI) continues to be one of the major causes of malnutrition in low and middle income countries (LMICs) [1]. The definition of FI is either as lack of nutritionally adequate and safe food or as limited ability to acquire food in socially acceptable ways [2]. It encompasses four hierarchical dimensions: availability, accessibility, utilization and stability [3,4]. “Availability” refers to sufficient quantities of quality food for an active and healthy life [4] and includes food production and distribution [5]. “Accessibility” refers to economic or physical resources for acquiring food [4] and includes affordability and preference [6]. “Utilization” refers to the intake of sufficient and safe food and includes the physical, social and human resources to transform food into meals [4], as well as societal values [6]. The dimension “stability” recognizes that FI can be transitory and cyclic or chronic [4] because of climate change, conflict, job loss, disease and other factors that can disrupt any one of the first three dimensions [7].

According to a report by FAO et al. [8] (p. 4), about 256.1 million people (20% of the total population) in Africa are undernourished; of these, 239.1 million live in sub-Saharan Africa. Within the sub-Saharan African region, the southern African countries continue to have the lowest burden of undernourishment (5.3 million), while eastern Africa has the highest burden in terms of numbers (133.1 million). In southern Africa, the latest UN World Food Programme (WFP)’s Hunger Map [9] puts South Africa, Namibia and Mauritius in the best position, with a prevalence of undernourishment between 5% and 14.9%, while Madagascar emerges as the country with the highest prevalence (>35%); no WFP data are available for Zimbabwe and Zambia. According to the report by FAO et al. [8] (p. 51), climate change, conflict and economic slowdowns and downturns are the major causes of FI in the region and its sub-regions. Poverty, socioeconomic disparities, social exclusion and rapid population growth also play a significant part in the rise in FI. At the household level, most research from LMICs gives poor extension services (e.g., lack of dissemination of relevant information needed to boost food production, particularly on farming methods and techniques including adequate funding and introduction of improved varieties of crops and breeds of animals), climate change, large family size, poverty (due to low income or unemployment) [10] and instability in government policies or poor governance as the major causes of FI [10,11]. Other causes include: lack of or inadequate access to productive resources, fluctuations in food availability and access, high food prices, diseases and pest infestations [10,12].

Research evidence from LMICs has found an association between FI and negative health outcomes. In children, FI has been associated with decreased academic performance, decreased emotional and intellectual development, delayed development of motor skills, poor general health, iron deficiency anaemia [13], underweight and obesity [1] and stunting [14]. In adults, it has been associated with diabetes, hypertension [15,16], anxiety, depression and suicide ideation [17,18] and other negative health outcomes [16,19], including intimate partner violence perpetration [20,21]. In Africa, a systematic review by Trudell et al. [22] found that FI was associated with poor mental health, depending on specific characteristics of study population. The authors of the review called for further research to include populations at risk to better understand the factors that could mediate the found relationship in order to inform policies and appropriate interventions [22]. Furthermore, in another review carried out in sub-Saharan Africa on the relationship between FI and key risk factors for diet-sensitive non-communicable diseases (NCDs), Nkambule et al. [23] found that FI was associated with dyslipidaemia, hypertension and overweight, especially among women. In the review, Nkambule and colleagues argued for the need to address FI as an integral component of diet-sensitive NCDs prevention programmes [23].

There is a dearth of systematic reviews on FI and health outcomes in southern Africa. For instance, a systematic review by Haines et al. [24] on risk factors for depression in people living with HIV (PLHIV) found that FI was one of the key risk factors, and thus FI needed to be considered when developing appropriate interventions to improve antiretroviral therapy (ART) outcomes [24].

Given the potential implications of FI for health outcomes, it becomes crucial to fully understand the major dimensions of FI as well as to know which instrument is most appropriate to measure FI and its dimensions [2]. Many researchers rely on the United States Department of Agriculture (USDA) Household Food Security Survey Module (HFSSM) for data on household FI. The HFSSM is an 18-item, self-report household measure of uncertain, insufficient or inadequate food access, availability and utilization, which assesses the food security (FS) situation of adults and children in a household over the past 12 months [2,7]. One limitation of this instrument is that it was designed to capture mostly financial constraints, poor resources and compromised eating patterns and consumption [7]. According to Webb et al. [7], however, the HFSSM, with some modification, is a valid instrument and has been widely used in countries across America. Based on the USDA instrument and the Latin American and Caribbean Food Security Scale (ELCSA), the FAO has developed and validated the Food Insecurity Experience Scale (FIES) for global use [25].

The burden of FI including its major determinants in some countries of the southern African region is not well known, and there is also a lack of systematic reviews. Therefore, this paper sought to review studies that empirically examined the relationship between FI and health outcomes among adults in southern Africa. The following question was sought: in which countries of the sub-region were the studies carried out and what were the most frequent health outcomes related with food insecurity?

## 2. Materials and Methods

### 2.1. Search Strategy

A systematic literature search was conducted using the Web of Science, PubMed and Google Scholar. The review was conducted and reported following the Preferred Reporting Items for Systematic Reviews and Meta Analyses (PRISMA) statement [26]. A PRISMA flow chart is presented in Figure 1. The review covered studies published from January 2001 to October 2021 in order to include studies carried out in the last two decades on FI and health outcomes. Studies were included if they were published in full-text English language, were empirical studies conducted in southern African countries and addressed the relationship between household FI and health outcomes. Search terms included “food insecurity OR food security OR household food insecurity AND health outcomes”, “food insecurity OR food security OR household food insecurity AND health-related outcomes” and “food insecurity OR food security OR household food insecurity AND health”. We excluded conference papers as well as conceptual, theory-building, review articles or articles that did not focus on household food insecurity.

### 2.2. Article Selection and Assessment

A total of 101 articles were identified in Web of Science, PubMed and Google Scholar. After duplicates were removed, 71 articles remained to be thoroughly screened. All 71 were retrieved and exported to Zotero reference management software [27] for manual screening, which was undertaken in two steps. In the first step, E.M. removed 46 articles (e.g., conceptual articles and reviews). In the second step, the full text of the remaining 25 articles was assessed for eligibility, and a further six articles were removed (e.g., water insecurity or child malnutrition without data on household FI). The remaining 19 articles were further assessed by G.M., and nine were removed (e.g., FI and medication adherence). Ten articles met the inclusion criteria. An additional search was performed to find empirical studies published between October 2020 and October 2021, which yielded four more articles for inclusion (see Figure 1).

## 3. Results

This section first presents the characteristics of the studies included in the review and then describes their findings.

### 3.1. Characteristics of the Included Studies

Fourteen studies published between 2006 and 2021 met the inclusion criteria. Five were conducted in South Africa [17,18,19,28,29], two in Zimbabwe [30,31] and one each in Zambia [32], Tanzania [33], Madagascar [34], Malawi [35], Namibia [36] and Botswana [37]. One study was carried out in Botswana and Swaziland [38]. Three studies used mixed methods, ten used quantitative methodology and one used qualitative methods including in-depth interviews, participant observations and focus group discussions. The sample sizes varied from 53 [31] to 8790 respondents [30] (see Table 1). We found no peer-reviewed studies that were carried out in Mozambique, Democratic Republic of Congo and Angola examining the relationship between FI and health outcomes.

### 3.2. Prevalence of Food Insecurity

In this review, most of the studies, except Farris et al. [34] and Muderedzi et al. [31], measured FI using validated instruments based on the HFSSM and the Household Food Insecurity Access Scale (HFIAS). The reported prevalence of FI across the countries varied from 18% in Botswana and South Africa [28,38] to 91% in South Africa [29] (see Figure 2).

Koyanagi et al. [19] in a South African study measured FI in the previous 12 months using two questions based on the USDA HFSSM. They found that 31.8% of the participants were food insecure. Of these, 11% had moderate FI and 20.8% had severe FI. Abrahams et al. [17], studying a population of pregnant South African women, examined FI in the last 6 months using six items from the HFSSM, and found that 42% were food insecure. Hadley and Patil [33] studying the situation in rural Tanzania, assessed FI in the last 3 months using 15 items from the USDA HFSSM, and found that 36% were food insecure.

McCoy et al. [30] measured FI in the last 4 weeks in a Zimbabwean population using three items from the HFIAS. They found that 51% were food insecure. Of these, 33% had moderate and 18% had severe FI. Dewing et al. [18], Wang et al. [37] and Mark et al. [35] measured FI in the last 4 weeks using nine items from the HFIAS and found that 59.8%, 46.67% and 49.7%, respectively, of the participants were food insecure. Kazembe et al. [36] assessed FI in the last 30 days using nine items from the HFIAS, as well as FI in the past 24 h using the Household Dietary Diversity Score (HDDS), and found 77.2% of the households were food insecure, but consumed less of unhealthy foods than those who were food secure (i.e., they consumed less of starch-oil-sugar diet and/or processed food).

Cole and Tembo [32] evaluated FI using the modified seven-item scale on coping strategies proposed by Maxwell [40] and found a prevalence of 64%. Of these, 17.5% had low FI, while 21.1% had moderate and 25.4% severe FI. Weiser et al. [38] estimated FI using one single question on food insufficiency in the last 12 months, and found that more women (32%) than men (22%) were food insecure. Naicker et al. [29] examined trends of FS in an urban South African population using a modified ten-item version of the Radimer/Cornell questionnaire. They found that in 2006, 85% of households were food insecure; in 2009, FI increased and peaked at 91%; and in 2012, it decreased to 70%. Jesson et al. [28], also in South Africa, measured FI over the past 30 days using three items from the Household Hunger Scale (HHS), and found that 18% of participants were food insecure.

In these studies, FI was associated with a variety of determinants, with a focus on poverty (due to unemployment, low income, lack of full-time employment or poor governance), high food prices, illness, dependency ratio, having three or more children, gender inequality, seasonality and food unavailability.

### 3.3. Food Insecurity and Associated Health Outcomes

The studies in the review addressed different dimensions of FI and how various structural factors within households or society affect these and produce different health outcomes. Most of the studies used a cross-sectional design, except the studies by Cole and Tembo [32] (longitudinal design) and Jesson et al. [28] (prospective cohort study).

Farris et al. [34] aimed to assess the drivers of food choices and barriers to diet diversity among a vulnerable population of caregivers in Madagascar. The findings revealed that though health concerns were consistently reported as one of the most important factors when shopping or preparing food items, they were not translated into actual dietary choices. Furthermore, FI was associated with poor general health, chronic malnutrition and poor academic performance in children [34]. Kazembe et al. [36] examined the association between dietary patterns and non-communicable diseases in Windhoek, Namibia. They found that higher levels of consumption of diets consisting largely of a starch, oil and sugar diet and/or processed foods were associated with increased odds of having hypertension, diabetes and cardiovascular disease, but that these products were consumed more by food-secure than by food-insecure households. Naicker et al. [29] evaluated the association between FI and health outcomes in an informal settlement in Johannesburg. They found that 24% of participants were affected by any kind of chronic disease (diabetes mellitus, hypertension or heart disease). Twenty percent of households with a member who had a chronic disease were food insecure. Thirteen percent of respondents in food-insecure households screened positive for common mental health disorders (anxiety or depression) compared with 1% in food-secure households. On the other hand, Koyanagi et al. [19] examined the association between FI and mild cognitive impairment (MCI) among middle-aged and older adults in South Africa. The authors found that FI was strongly associated with MCI, particularly among individuals aged ≥65 years.

The study by Muderedzi et al. [31] explored the relationship between FI, gender roles and human immunodeficiency virus (HIV)/acquired immune deficiency syndrome (AIDS) among caregivers and health care personnel in Binga, one of Zimbabwe’s poorest and most food insecure districts. The findings showed FI, gender roles and HIV/AIDS entwined in a vicious cycle that heightens vulnerability to and worsens the severity of each condition. McCoy et al. [30] examined the association between FI and women’s uptake of services to prevent mother-to-child HIV transmission (MTCT) in Zimbabwe. The findings showed that FI was not associated with maternal or infant antiretroviral therapy (ART)/ARV prophylaxis. However, among HIV-exposed infants, 13.3% were born to women who reported severe household FI, compared with 8.2% of infants whose mothers reported FS. Among pregnant and postpartum women, FI was inversely associated with use of prevention of MTCT (PMTCT). Weiser et al. [38], in turn, examined the association between FI and risky sexual behaviours in Botswana and Swaziland and found food insufficiency, especially among women, was associated with inconsistent condom use with a non-primary partner, selling sex, intergenerational sexual relationships and lack of control in sexual relationships. In contrast, Wang et al. [37] evaluated the association between FI and HIV infection with depression and anxiety among tuberculosis (TB) patients in Gaborone, Botswana. They found that food-insecure participants had 2.3 and 1.4 times the odds of experiencing depression and anxiety, regardless of being HIV-positive or not.

With regard to mental health, Abrahams et al. [17] analysed factors associated with FI and depression among pregnant women from a low-income suburb in Cape Town. They found the odds for FI were increased in women with suicidal behaviour (odds ratio (OR = 5.34)) and depression (OR = 4.27) and in those with three or more children (OR = 3.79). In addition, FI was strongly associated with major depressive episodes (MDEs). Hadley and Patil [33] examined the association between FI and maternal anxiety and depression among four ethnic groups living in two communities of rural Tanzania. The findings showed a strong positive correlation between a caretaker’s FI score and her summed response on the Hopkins Symptom Checklist (HSCL) (*p* < 0.0001). High maternal age (*p* = 0.029) and household FI (*p* < 0.0001) were significant predictors of HSCL scores. The study by Cole and Tembo [32] examined the association between FI and self-reported mental health in rural Zambia. Their findings showed a positive and significant association between FI and poor mental health (r = 0.39, *p*-value < 0.0001). In addition, FI during the dry season, the time of relative food abundance, had greater effect on mental health than FI during the rainy season. Likewise, Mark et al. [35] examined the association between FI and clinical depression and how seasonality modifies this relationship among post-partum women in the rural Ntcheu District of Central Malawi. They found participants reporting higher levels of FI had 4.6 times the odds of meeting the cut-off for clinical depression compared with those reporting lower levels of FI, and the effect of FI on mental health was greater during the dry season (OR = 9.9) than during the rainy season (OR = 2.6). Dewing et al. [18], in turn, investigated the associations between FI, postnatal depression, hazardous drinking and suicide risk among women in the outskirts of Cape Town. In their study, 79 (31.7%) women met screening criteria for probable depression, and 39 (15.7%) met screening criteria for hazardous drinking. Nineteen (7.6%) women had significant suicidality, seven (2.8%) of whom were classified as high risk. Food insecurity was strongly associated with postnatal depression, hazardous drinking and suicidality. Lastly, Jesson et al. [28] assessed the association between FI and depression among youth living in Durban and Soweto, South Africa. They found those individuals with probable depression had increased odds (OR = 2.79) of being food insecure compared with those who were not depressed.

## 4. Discussion

### 4.1. Prevalence of Food Insecurity

The majority of the studies included in this review used quantitative and mixed methods, and the main findings used quantitative methodology. The same pattern in terms of methods was found among empirical studies that examined FI in other regions of sub-Saharan Africa [41,42,43].

In the current review, the prevalence of FI across the countries varied from 18% in Botswana and South Africa [28,38] to 91% in another study from South Africa [29]. The major factors associated with FI were: poverty, illnesses (mostly HIV and tuberculosis), high food prices, dependency ratio, having three or more children, gender inequality, seasonality and food unavailability. The variations in FI across the countries may be due to the measuring instruments, the specific characteristics of the study populations and the time period used for data collection [44]. Nonetheless, most of the studies used reliable and valid instruments for measuring FI, in most cases, modified versions of the USDA HFSSM and the HFIAS [45,46,47].

Food insecurity in Africa is largely associated with poverty [48]. Many factors can contribute to poverty, but flawed economic policies and institutions suffering from corruption, poor governance and political conflicts [49,50] as well as poor land utilization are considered the primary causes of poverty in Africa [49]. Hence, effective interventions from governments, the private sector and international institutions such as the World Bank and International Monetary Fund are needed to stimulate decent work and build inclusive economic growth [49,50]. In addition, low per capita income and rapid population growth make it extremely difficult to have savings and invest in Africa, a condition that perpetuates low productivity and low income [50].

Moreover, the majority of poor urban households purchase the bulk of their food rather than produce it themselves, and therefore, household income and food prices are critical determinants of FI in cities [48]. Food insecurity tends to exacerbate with illness, high food prices observed in cities, dependency ratio and having three or more children, as highlighted in this review. For instance, various studies indicate that the COVID-19 pandemic has caused a rise in FI [51] through food shortages and high food prices, job losses and decrease in livelihoods [52,53]. In a unique way, the COVID-19 pandemic highlighted the fragilities of health systems and food systems not only in LMICs [54,55], but also in high income countries [56,57]. A study by Rosenberg et al. [58] on the impact of the COVID-19 pandemic on labour markets and economies of 16 southern African countries found that the COVID-19 pandemic was associated with increased job loss risk specifically in Angola, South Africa and Zimbabwe. These studies suggested the need to provide social and economic support to the most vulnerable groups [59,60,61] as well as the need to rethink future actions towards global food security, since COVID-19 greatly affected all its four pillars [56,59]. Furthermore, FI in urban areas is largely an outcome of income poverty often accompanied by lack of secure employment, but also includes living conditions and inconsistent access to water, sanitation, electricity and other resources that shape households’ food utilization capacity [62].

On the other hand, there is a body of evidence about the relationship between household FI and gender inequality in Africa and other LMICs, and this suggests that socioeconomic interventions are needed to empower women and close the FI gap between female- and male-headed households [63,64]. In contrast to the cities, in rural areas, factors such as seasonality and climate change can have a huge impact, as most households rely on their own food production. Smallholder farmers cannot afford the cost of fertilizers, high quality seeds and planting materials, especially the cost of irrigation, nor can they afford quality education, and therefore, depend on rainfall, which is becoming increasingly unreliable because of climate change [65].

With regard to factors associated with FI, similar results have been reported for Africa and elsewhere, particularly in LMICs [66,67,68]. Besides poverty, climate change [66,67], rapid population growth [11], poor governance and political conflicts [11,68] have been highlighted as relevant determinants of FI. For instance, climate change may affect food systems and FI in a number of ways ranging from direct effects on food production (e.g., variations in rainfall patterns leading to droughts or floods, vacillations in temperatures and variation in the length of growing seasons) to changes in markets, food prices, increased poverty and food supply chain infrastructure [6,67]. The quality of governance (e.g., government effectiveness, political stability, rule of law, accountability) and targeted policies are critical for fostering an adequate environment, which is vital to all economic processes and investments, especially those related to enhancing national food and nutrition security, social protection and the pace of economic growth [11].

### 4.2. Food Insecurity and Associated Health Outcomes

The studies included in this review examined the relationship between FI and health outcomes in adults. For instance, FI in South Africa was associated with MCI [19], depression and anxiety [17,28], postnatal depression, hazardous drinking and suicide risk [18], chronic diseases (diabetes mellitus, hypertension and CVD) [29] and perceived stress [28].

Koyanagi and colleagues [19] found that the odds of having MCI were 3.87 times higher in individuals aged ≥65 years with severe FI than their food-secure counterparts. Their findings are supported by other studies [69,70]. The mechanisms linking FI to MCI are not clear, but it is believed that stress derived from FI can increase the risk of cognitive decline [71]. Moreover, FI often compromises diet quality, and poor diet has been associated with higher risk of cognitive decline [72,73].

Abrahams et al. [17] found that the odds of experiencing depression were five times greater in food-insecure compared with food-secure women. They also found that the odds of being food insecure were 4.3 times greater in depressed women compared with women who were not depressed. In fact, many studies conducted in LMICs suggest that FI and common mental illnesses are related in a vicious cycle, but the strength of this association depends on various factors, including the specific characteristics of the study population, the measuring instrument used and the time period for data collection. For instance, in Ecuador, self-reported depression in mothers increased the odds of being food insecure by almost three times [16], and in Korea it was by four times [74]. On the other hand, the study by Cole and Tembo [32] in rural Zambia revealed that FI during the dry season had greater effects on mental health than FI during the rainy season. Along the same lines, Mark et al. [35], in rural Malawi, found FI during the dry season to be associated with greater odds of experiencing clinical depression (OR = 9.9) than FI during the rainy season (OR = 2.6). One plausible explanation is that, because the dry season is supposed to be the time of relative food abundance and FS, therefore, when households fail to harvest an adequate amount of food, they are forced to rely on various coping strategies, and this becomes the source of worry and anxiety [32,35]. Likewise, the rainy season is associated with disease outbreak, which brings uncertainty about physical health and future food production [32].

Overall, across investigated countries, FI was associated with chronic malnutrition, poor general health and poor academic performance [34], anxiety and depression [33,37], poor mental health [32] and increased odds of hypertension, diabetes and CVD [36]. Many studies have shown that food-insecure children have difficulty in achieving academic success [75], and FI has been associated with poor cognitive and emotional development [76]. Other findings include those by Farris and colleagues [34] who found that, though health concerns were consistently reported as relevant when buying food, they were not translated into actual food choices. For instance, green leafy vegetables were not viewed as nutrient-dense because of their relatively lower cost and confusion over energy-dense vs. nutrient-dense food [34]. Hence, educational initiatives are needed to educate the population about the nutritional value and benefits of consuming green leafy vegetables.

Kazembe et al. [36] found that “starch–oil–sugars” and “meat–fish–dairy” dietary patterns were associated with hypertension, diabetes and CVDs, but that these products were less consumed by food-insecure households. This is an unusual finding that food-insecure households were found to consume less of unhealthy foods than their counterpart who were food secure (i.e., they consumed less of the starch–oil–sugar diet and/or processed food). The authors of the study explained their findings, pointing out that, on one hand, there was a lack of processed foods in informal markets and, on the other, food-insecure households could not afford products such as meat, and only managed to buy them for festive occasions. Another explanation is the role of food transfers (e.g., grains, vegetables and wild foods) in mitigating FI and combating NCDs in informal settlements, as highlighted by the authors. The HDDS instrument, which was used by Kazembe et al. [36], captures the types of food the households have consumed in the last 24 h, hence it is possible that it captures chance findings, and the findings depend on the season when the instrument is administered. The availability of food transfers for food-insecure households in informal settlements is dependent on the season. Moreover, this instrument is recognized in Namibia as inadequate for capturing, for instance, the consumption of wild foods (which are very healthy), as is common in informal settlements [77,78]. With regard to the negative health outcomes revealed, there is a body of evidence that a dietary pattern characterized by high intake of vegetables, fruit and complex carbohydrates and low intake of processed meat and refined carbohydrates may have a protective effect against hypertension, type 2 diabetes, CVDs and other NCDs [79]. These findings require further investigation, as it has been reported elsewhere that food-insecure households are more likely to buy cheap and unhealthy food (e.g., highly processed foods with large amount of sugar, sodium and oils), and therefore have increased risk for NCDs [15,80].

According to Davison et al. [81], the mechanisms linking FI to mental illnesses include biological mechanisms, by which micronutrient deficiency impacts mental health, and psychological mechanisms, by which stress, worry and anxiety that derive from FI may cause maladaptive responses, leading to suicidal ideation and behaviour. Furthermore, suicidal behaviour, as with other mental illnesses, may lead to increased health expenses, unemployment, social withdrawal and other consequences that worsen each of these conditions [82].

In Zimbabwe, FI was associated with increased risk of HIV acquisition [30,31], and severe FI was considered a barrier to uptake PMTCT services [30]. Muderedzi et al. [31] illustrate the relationship between FI, gender roles and HIV/AIDS acquisition as a vicious cycle. These authors found that the four dimensions of FS were missing in Tonga communities because of extreme poverty. For this reason, women, even if aware of the related health issues, often adopted sexual (as well as non-sexual) coping strategies to obtain food for their families. Most women in their study had taken numerous HIV tests and were eager to get an HIV-positive result because several NGOs in the area provide food aid to HIV-positive people. These women often stated that HIV came with food, and that the need for food was far more important than getting HIV. In addition, the authors stated that Tonga cultural factors (e.g., polygamous marriage systems, widow inheritance and sexual cleansing) are important factors in the spread of HIV [31].

Along the same lines, Weiser et al. [38] found that FI was associated with risky sexual practices among women in Botswana and Swaziland. These findings suggest that FS could play an important role in limiting the spread of HIV/AIDS in sub-Saharan Africa. Furthermore, income generation programmes together with educational initiatives and interventions on food supplementation and food production could alleviate gender disparities that drive risky sexual practices, and therefore reduce HIV/AIDS acquisition among high-risk populations [38]. Most of the studies reporting associations between FI and HIV in Africa have focused on the effects of FI on ART and have found that approaches aiming to enhance the FS of people living with HIV (PLHIVs) may be effective for improving HIV outcomes [83,84,85]. For instance, Benzekri et al. [83] aimed to assess the retention on ART and to identify the predictors of loss to follow-up (LTFU) among PLHIVs in Senegal. They found that severe FI (OR = 2.55) was the strongest predictor of LTFU, and therefore, implementing strategies to enhance FS for PLHIV may be effective for reducing LTFU and strengthening the HIV care cascade in Africa. Similar results have been found in Democratic Republic of Congo [84] and Mozambique [85,86]. A systematic study by Singer et al. [87] in Africa, Europe, Canada, USA and South America found that FI was an important barrier to ART adherence, and food assistance appeared to be a promising intervention strategy to improve ART outcomes among PLHIV. Thus, household FS or food assistance programmes should be an essential component of HIV/AIDS strategies to improve ART outcomes [87,88]. On the other hand, a study by Nagata et al. [89] using nationally representative data on US young adults aged 24–32 years has found that food-insecure young women had greater odds of self-reported sexually transmitted infections (STIs), particularly chlamydia and HIV, exchanging sex for money and having multiple concurrent sex partners in the past 12 months compared with food-secure ones. In both young men and women, FI was associated with greater odds of substance use. These findings have public health and clinical implications regarding alleviation of FI and improved health outcomes.

It is noteworthy that for Mozambique and Democratic Republic of Congo, no studies were found examining the association between FI and health outcomes although FI is a living reality for many households [90,91,92,93,94]. In Mozambique and Democratic Republic of Congo, the studies found addressed the effects of FI or food assistance programmes on ART, as indicated in the previous paragraph [84,85,86,87]. In Angola, one study examined malnutrition in children under 2 years old and its associated factors [95]. Unfortunately, some potential determinants of child malnutrition (e.g., household FI, sanitation, primary care) were not measured, and diarrhoea (PR 1.39 [95% CI: 1.01–1.87]) was associated with stunting (32% [95% CI: 28.7–35.5]). Nonetheless, Humbwavali and colleagues [95] recognized that they identified few risk factors for child malnutrition, and suggested that a combination of life course factors related to pregnancy and birth, and collective exposures (e.g., household FI, sanitation, primary care, infectious diseases) could be causing child malnutrition in Angola. In addition, the authors argued that probably because of pre-existing nutritional deficits, these children were more vulnerable to infections that led to diarrhoea or these children had repeated episodes of diarrhoea that led to stunting. In any case, these conditions were entwined in a vicious circle which strengthened each condition [95]. Thus, joint and coordinated efforts between governments, private sector, non-government organizations and communities are needed to improve child malnutrition in the region in particular and in Africa as a whole [95,96,97,98,99]. On the other hand, a recent case-control study by Robbiati et al. [100] found an association between severe FI and diabetes among adults in the capital city of Angola. Therefore, the authors called for further longitudinal studies to assess the pathways linking food insecurity and diabetes in that setting.

### 4.3. Areas for Further Research

This review has shown that FI is an important societal and public health problem in southern Africa, and it has been measured using different instruments, but mostly the USDA HFSSM and the HFIAS. Food insecurity has been associated with a variety of factors, mainly poverty, illness, high food prices, dependency ratio, gender inequality, seasonality and food unavailability. Crush and Frayne [48] argue that FI in Africa is largely the result of poverty and that many factors may contribute to poverty. In other research, flawed economic policies and institutions (e.g., corruption, poor governance, political conflicts and poor land utilization) are considered the primary causes of poverty in Africa [49,50]. Therefore, future studies that investigate the role by structural and institutional policy on FI are warranted. In addition, the studies included in this review indicate that FI has detrimental consequences for the physical and mental wellbeing of those affected; in adults, FI has been associated with poor mental health, anxiety and depression, type II diabetes, hypertension and other negative health outcomes. These findings suggest an urgent need for research that can also provide a deeper understanding of the determinants of FI in southern African countries for which empirical research is still not available (e.g., Mozambique, Democratic Republic of Congo). Such research will help to disentangle mechanistic pathways of the relationship between FI and physical and psychological health outcomes across the region.

### 4.4. Strengths and Limitations

This review provides one of the first assessments of empirical studies that examined FI and its relationship with health outcomes in southern Africa. Broad search terms were used to identify as many articles as possible on the topic. Moreover, the selection process was conducted in two stages to increase the number of included papers. Nonetheless, the review has some limitations that need to be acknowledged. It relied exclusively on peer-reviewed studies published in English in databases. Therefore, it is very likely that other publications addressing FI and health outcomes written in other languages do exist, but that they were not included.

## 5. Conclusions

This review investigated empirical evidence of the relationship between FI and health outcomes in southern Africa and found 14 studies carried out in South Africa, Zimbabwe, Zambia, Tanzania, Madagascar, Malawi, Namibia, Botswana and Swaziland. No studies were found on this subject in Mozambique and Democratic Republic of Congo. The majority of the identified studies used quantitative methodology, which is in line with reviews conducted in other regions of sub-Saharan Africa. A variety of instruments were used to collect information about FI, with most studies using the USDA HFSSM and the HFIAS. Food insecurity in this sub-region was associated with a number of factors, but poverty, illness, high food prices, dependency ratio, having three or more children, gender inequality, seasonality and food unavailability were consistently found to be the main drivers of FI. Furthermore, FI was associated with poor mental health, anxiety and depression, type II diabetes, hypertension, CVD and increased risk of HIV acquisition through biological, psychological and behavioural mechanisms. These findings suggest that future research is needed to identify and standardize measures of FI suitable for the southern African context and inform public health policies and design appropriate interventions to alleviate household FI and improve health outcomes. This is crucial for countries such as Mozambique and Democratic Republic of Congo for which no current studies exist that address the relationship between food insecurity and health outcomes.

## Figures and Tables

**Figure 1 ijerph-19-05082-f001:**
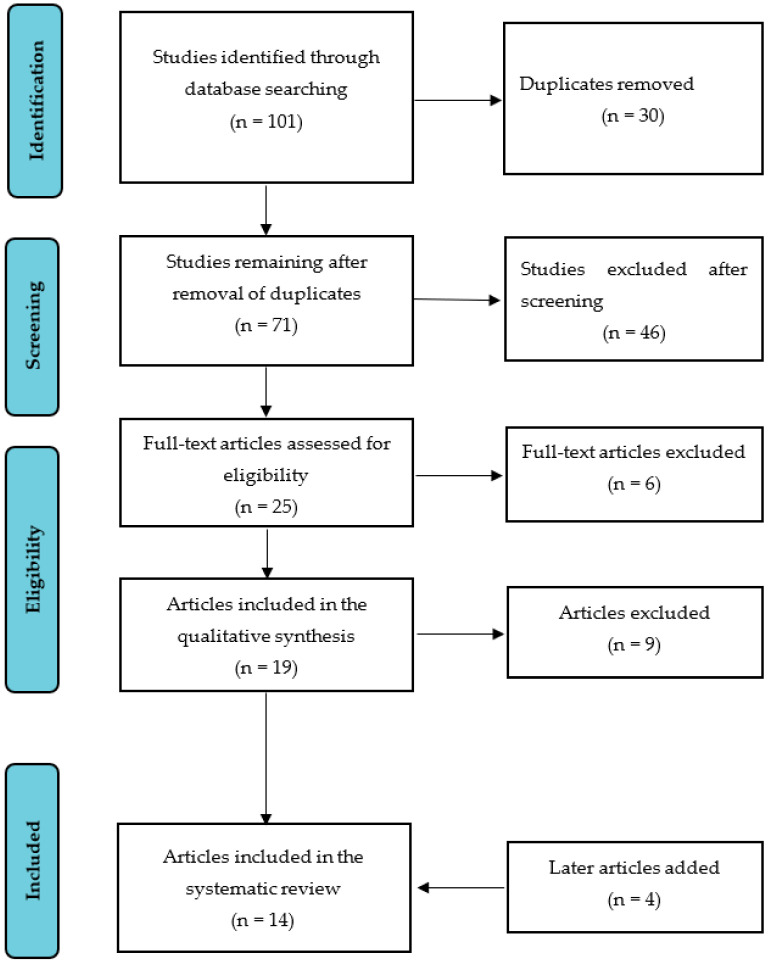
Preferred Reporting Items for Systematic Reviews and Meta Analyses (PRISMA) flow chart of the literature search and selection process.

**Figure 2 ijerph-19-05082-f002:**
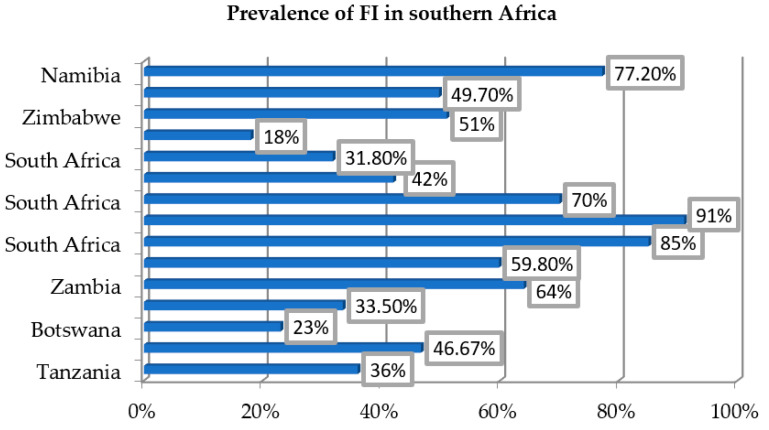
The prevalence of FI across southern African countries included in this systematic review.

**Table 1 ijerph-19-05082-t001:** Summary of the 14 studies included in the systematic review.

Author, Year/Country/Reference	Study Objective	Design, Sample, and Methods of Analysis	Measurement of FI	Prevalence of FI	Factors a Associated with FI	Associated Health Outcomes	Findings (Overall)
Hadley and Patil, 2006/Tanzania/[33]	To examine whether FI is associated with anxiety and depression	Cross-sectional study (survey and ethnography), *n* = 449, Logistic regression	15 items from the USDA HFSSM to measure FI in the last 3 months	36% were food insecure	Poverty, ethnicity, large household size	Maternal anxiety and depression	There was a strong positive association between FI and maternal anxiety and depression
Weiser et al., 2007/Botswana and Swaziland/[38]	To examine the association between FI and risky sexual behaviours; to determine whether gender modifies this association	Cross-sectional study, *n* = 2051 (1255 in Botswana and 796 in Swaziland), Logistic regression	One single question on food insufficiency in the last 12 months	32% of women were food insecure (Botswana 28%, Swaziland 38%) vs. 22% of men (Botswana 18%, Swaziland 29%)	Poverty, socioeconomic differences, gender inequality and unequal household food allocation	Multiple risky sexual practices among women (selling sex, and engaging in unprotected sex and intergenerational sex)	Food insufficiency was associated with inconsistent condom use with a non-primary partner, sex exchange, intergenerational sexual relationships and lack of sexual control
Cole and Tembo, 2011/Zambia/[32]	To examine the association between FI and self-reported mental health	Longitudinal design (surveys and ethnography), *n* = 280, Multilevel regression model	Modified 7-item scale on coping strategies	64% were food insecure (17.5% low, 21.1% medium, 25.4% high)	Seasonality, dependency ratio and illness	Poor mental health	There was a positive and significant association between household FI and poor mental health
Dewing et al., 2013/South Africa/[18]	To assess the relationship between FI, postnatal depression, hazardous drinking, and suicidality	Cross-sectional study, *n* = 249, Poisson regression models	9 items from the HFIAS to measure FI in the last 4 weeks	59.8% were severely food insecure	Unemployment or very low income, HIV endemic and tuberculosis	31.7% postnatal depression, 15.7% hazardous drinking, and 7.6% suicide risk	FI was strongly associated with postnatal depression, hazardous drinking and suicide risk
Naicker et al., 2015/South Africa/[29]	To assess trends in FS from 2006 to 2012; to determine the main predictors of FI and its associated health outcomes	Cross-sectional studies, *n* = 188, Logistic regression	Modified 10-item version of the Radimer/Cornell Questionnaire on household FS	In 2006, 85%, in 2009, 91%, and in 2012, 70% of households were food insecure	Low asset ownership, having major residence problems, having a very low income or no full-time employment	Chronic disease (diabetes mellitus, hypertension or heart disease); 13% common mental disorders (anxiety or depression)	Household FI decreased from 2006 to 2012, though there was a spike in 2009, and FI was translated to poor food diversity
McCoy et al., 2015/Zimbabwe/[30]	To examine the association between FI and women’s uptake of services to prevent MTCT	Cross-sectional study, *n* = 8790, Poisson regression models	3 items from the HFIAS to measure FI in the last 4 weeks	51% were food insecure (33% moderately, 18% severely)	Poverty, province of residence, being single, low education and large household size	Increased risk of HIV acquisition	Severe FI is a barrier to uptake of some PMTCT services, contributing to HIV transmission from mother to child
Abrahams et al., 2018/South Africa/[17]	To assess factors associated with FI and depression and anxiety	Cross-sectional study, *n* = 376, Logistic regression	6 items from the USDA HFSSM to measure FI in the last 6 months	42% were food insecure	Poverty, unemployment, having three or more children, pregnancy	21.4% were depressed (MDE); 22.8% had anxiety disorder	The odds of being food insecure were increased in women with suicidal behaviour (OR = 5.34), depression (4.27) and three or more children (3.79)
Muderedzi et al., 2019/Zimbabwe/[31]	To explore the relationship between FI, gender roles and HIV/AIDS	Qualitative study (in-depth interviews and focus group discussions), *n* = 57, Thematic analysis	Not available	Not available	Extreme poverty, poor governance, lack of food, gender inequality, transactional sex	Increased risk of HIV acquisition	Most women were eager for an HIV-positive result in order to secure food handouts from NGOs to feed their families
Koyanagi et al., 2019/South Africa/[19]	To assess the association between FI and mild cognitive impairment (MCI)	Cross-sectional study, *n* = 3672, Logistic Regression	2 questions on FI in the last 12 months based on the USDA HFSSM	31.8% were food insecure (11% moderately, 20.8% severely)	Extreme poverty, race, low education, HIV endemic, high food prices	MCI (8.5%)	Moderate and severe FI was associated with 2.82 and 2.51 times the odds of having MCI
Farris et al., 2020/ Madagascar/[34]	To assess the drivers of food choices and barriers to diet diversity	Cross-sectional study (survey and focus group discussions), *n* = 137, Descriptive statistics and thematic analysis	5 questions on FI as described by Rakotosamimanana et al. [39]	Not available	Poverty, high food prices, lack of food and food beliefs	Poor general health, chronic malnutrition, poor academic performance	Though health concerns were consistently reported to be one of the major drivers of food choices, they were not translated into actual dietary choices
Wang et al., 2020/Botswana/[37]	To examine the association between FI and HIV infection, and depression and anxiety	Cross-sectional study, *n* = 180, Poisson regression models	9 items from the HFIAS to measure FI in the last 4 weeks	46.67% were food insecure (8.33% had mild FI; 9.44% had moderate FI; 28.89% had severe FI)	Poverty, co-infection of HIV and tuberculosis	Depression and anxiety	FI was associated with higher odds for depression (OR = 2.3) and anxiety (OR = 1.41) irrespective of being HIV-positive or not
Mark et al., 2021/Malawi/[35]	To evaluate the association between FI and clinical depression, and the modifying effects of seasonality	Cross-sectional study, *n* = 175, Logistic regression	9 items from the HFIAS to measure FI in the last 4 weeks	49.7% were food insecure	Poverty, having three or more children, seasonality, climate change	Clinical depression	Food-insecure participants had 4.6 times the odds of experiencing clinical depression, and the effect was greater in the dry season (OR = 9.9) than in the rainy season (OR = 2.6)
Kazembe et al., 2021/Namibia/[36]	To investigate the association between dietary patterns and non-communicable diseases	Cross-sectional study, *n* = 863, Factor analysis and logistic regression	9 items from the HFIAS to measure FI in the last 30 days and the HDDS to examine FI in the last 24 h	77.2% were food insecure	Poverty, low-income, informal settlements, type of housing, informal work	Increased odds of hypertension, diabetes and CVD	Food-insecure households less often had starch-oil-sugar diets and/or processed foods which are associated with increased odds of hypertension, diabetes and CVD
Jesson et al., 2021/South Africa/[28]	To assess the association between FI and depression	Prospective cohort study, *n* = 422, Logistic regression models	3 items from the HHS to measure FI in the last 30 days	18% were food insecure (12% had moderate FI and 6% severe FI)	Poverty, female-headed household, household size, not currently being a student, transactional sex	42% probable depression, 72% perceived stress, 63% probable anxiety	Participants with probable depression had increased odds (OR = 2.79) of being food insecure

United States Department of Agriculture (USDA); Household Food Security Survey Module (HFSSM); Household Food Insecurity Access Scale (HFIAS); Household Dietary Diversity Score (HDDS); Household Hunger Scale (HHS); Human immunodeficiency virus (HIV); Acquired immune deficiency syndrome (AIDS); Mild cognitive impairment (MCI); Cardiovascular disease (CVD); Major depressive episodes (MDEs); Prevention of mother-to-child HIV transmission (PMTCT); Non-governmental organizations (NGOs).

## Data Availability

The review used existing research data.

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
