# Peer review of "Food Insecurity and Health Outcomes Other than Malnutrition in Southern Africa: A Descriptive Systematic Review"

_ijerph, 2022, doi:10.3390/ijerph19095082_

Round 1

Reviewer 1 Report

Article could be accepted in present form.

Author Response

According to the first reviewer, the manuscript could be accepted in its present format. Thank you very much for your consideration.

Reviewer 2 Report

The authors have engaged in a second attempt to review health outcomes from food insecurity and submitted it to the biosafety section if the international journal of environmental research and public health. 

The title “Food Insecurity and Health Outcomes in Southern Africa: Descriptive Systematic Review” remains unchanged by the authors, and is supposed to cover a scope, which far exceeds the content of the paper. To put it simply, the main health outcomes from malnutrition such as child survival in the context of high malaria, diarrhea and pneumonia prevalences, developmental issues (cognitive and growth impairment),  adverse health consequences of nutritional deficiencies and exposure to foodborne hazards are not covered in the paper.

Consequently this paper is misleading, unless the title and scope are corrected. A correct title for this paper could be  Food Insecurity and Health Outcomes Other Than Malnutrition in Southern Africa: a Descriptive Systematic Review

The review does not explore the root cause of the selected health outcomes (beyond malnutrition) and whether there is a causality relationship between food insecurity and of some of the health outcomes.

Therefore the authors should be more self-critic about the empirical evidence that food insecurity is or could be correlated with the selected health outcomes.

In other words, I think that the authors should emphasize that they are not able to say to which extent the management of the selected health outcomes requires improving food security or increasing education or improving the health system, and which one seems more cost effective interventions.

Author Response

According to the second reviewer, we would like to clarify that this manuscript was submitted to Global Health section and not Biosafety as indicated by the reviewer.

About the comment that the manuscript is misleading, unless the title and scope are corrected, and the possible solution suggested by the reviewer, we see no problem as the suggested title is in accordance with our manuscript. Our first thoughts were that since we made it clear that we sought to examine the relationship between food insecurity and health outcomes among adults in southern Africa (see page 3), one could implicitly understand that malnutrition is not our main focus as it is often related to children. Therefore, we did not feel compelled to add it in the title.

About the comment that the manuscript does not explore the root causes of presented health outcomes beyond malnutrition, and whether there is a causality relationship between food insecurity and some health outcomes, our understanding is that this was not our focus. We presented and discussed a number of potential major causes of food insecurity in the region. On the other hand, most of the selected articles used cross-sectional design (see pages 6–9 and 11), and therefore, we could not claim a causality relationship between food insecurity and these negative health outcomes. These empirical studies have pointed out that food insecurity could be an important risk factor for a range of negative health outcomes, hence it has been suggested that food assistance programmes any programme designed to alleviate household food insecurity could eventually improve physical and mental health outcomes. In addition, it was pointed out that food insecurity could be linked to these negative health outcomes through biological, psychological and behavioural pathways.

About the comment that we should be more self-critics regarding the empirical evidence that food insecurity is or could be correlated with the presented health outcomes, we feel that we addressed this issue, but more studies, especially longitudinal ones are needed to ascertain causality effect.

With regard to the comment that we should emphasize the inability to tell to which extent the management of health outcomes could require improving food security or health system or increasing education, and which intervention seems more cost effective, we definitely agree that we couldn’t tell to which extent the management of specific health outcomes requires improvement of food security or health system or increasing education, but we made it clear that food insecurity is a multi-faceted issue, and therefore, joint efforts from government, private sector, civil society and communities are needed. Anyway, the interventions must be contextual to each case, group or specific country. For instance, poverty was one of key causes of food insecurity across the region. For that reason, efforts towards improving people’s education, improving household food security, creating more jobs and improving the health system could be more appropriate as medium and long-term strategies. On the other hand, food assistance programmes could be important as a short-term strategy for most vulnerable groups as indicated in the manuscript.

Reviewer 3 Report

The manuscript by Militao, E. et. al. entitled “Food Insecurity and Health Outcomes in Southern Africa: A Systematic Descriptive Review” is a review study with an aim to determine the impact food insecurity (FI) on health outcomes among people living in southern Africa region.

The review included 28 papers/studies.

Some technical comments:

English proofreading is suggested and repetitive sentences should be removed.

One question that need explanation, why Mozambique was chosen as an important area of interest? How many areas of southern Africa region are/aren’t covered by FI surveys?

Row 47/56 – p.4/ p.51 - report that you list in references has only 4 pages

Row 93 – already asked question why Mozambique

Row 146 – “in the last 3 months” – define exact time interval – is this last three months from now or last three months in paper or something else?

Row 120-121 - why did you subsequently added those references? If the answer is that it is because to complete 20 years, why this time interval was not set from the beginning of selection process?

Hypertension, Diabetes, Anxiety, depression and HIV

Part about FI  - one should comment not only results obtained in separate papers but also put results obtained in those papers in the context of the years when they were collected.

In the part where you connect FI with health outcomes try to group similar outcomes together

Row 277-279 – reference missing

Row 298-301 – that is not only situation in Africa but in all middle-income countries around the world.

Row 303 – LMIC (reference)

Row 422-425 – reference missing

You need to give more emphasis on FI on health outcomes in the manuscript because it is set as the main goal of this paper and more attention has been paid to socioeconomic reasons for their FI.

Author Response

According to the third and last reviewer, some technical comments were suggested. English proofreading was suggested and repetitive sentences should be removed. The entire manuscript was edited for language by professional services.

About the question why Mozambique and how many areas were covered by our manuscript, we would like to emphasize that southern Africa region (SADC) consists of 16 countries; of which, 3 countries are simultaneously primary members of eastern Africa (Tanzania) and central Africa (Angola and Democratic Republic of Congo). Our search included all the 16 countries, but in 9 countries we found empirical studies that met our research criteria. In some cases like Mozambique, Angola and Democratic republic of Congo, the empirical studies we found did not meet our main objective, but their results were worth mentioning (see pages 4, 5, 15 and 16). This manuscript is part of a larger project run in Mozambique, and we are associated with a number of follow up empirical studies about food insecurity and specific health outcomes in Mozambique. Anyhow, it is much clear now that it was not just the case of Mozambique, but also Angola and Democratic Republic of Congo as indicated before.

The report by FAO mentioned on page 2 has 104 pages and not 4 pages as suggested. This is modified in the current version of the manuscript (see page 17).

About the comment on the term “in the last 3 months”, it has to be understood that this is related to the instrument used to collect data on food insecurity (HFSSM). In this case, in the last 3 months refers to the time period the participants should recall their experiences. For instance, let us suppose we approached an adult on April 1st, this means one should recall his experiences of food insecurity from January 1st to March 31st.

The time period in this manuscript was already set from the beginning of selection process. It is just that the additional search was done to make sure that we didn’t left recent empirical studies conducted in southern Africa on this matter.

Hypertension, diabetes, anxiety, depression and increased risk of HIV acquisition were the health outcomes frequently associated with food insecurity. It is worth mentioning that our search was not confined just to these health outcomes.

About the comment on discussing the results also in accordance with the time period when the data was collected, we think this has been followed throughout the discussion section.

For sure, the results about FI and associated health outcomes were presented as suggested whereby similar outcomes were described together (see pages 11 and 12).

All the comments on missing references had already been fulfilled (see pages 12–15).

About the comment on smallholder farmers, we agree that what is presented might be true across LMICs. This is presented as a general statement, but the reference we brought was specific to Africa (see page 13).

It is definitely much clear now that we give more emphasis on FI and associated health outcomes as requested (see pages 13-16).

Round 2

Reviewer 2 Report

I am glad you corrected the title, which was important not not mislead the readers on its content. Best wishes

Author Response

The second reviewer (R2) made one comment of appreciation. The reviewer is glad that we corrected the title so that it is not misleading the readers. Thank you very much for your appreciation.

Reviewer 3 Report

The manuscript by Militao, E. et. al. entitled “Food Insecurity and Health Outcomes Other Than Malutrition in Southern Africa: A Systematic Descriptive Review” is a review study with an aim to determine the impact food insecurity (FI) on health outcomes among people living in southern Africa region.

I would like to thank the authors for their efforts to improve the manuscript. Some rearrangements have been made to the text.

Among others the title was slightly changed. Whether the title change relates to undernutrition or you also meant malnutrition that involves excessive food intake but insufficient nutrient density what can be cause of chronical diseases? (Because we know that malnutrition is associated not only with reduced BMI but also with obesity)

Row 110-11 – can you specify how many articles are from which database

Row 134 – how many articles were not eligible in additional search?

Reference 36 obtained very unusual results, so it is important to explain these results well at every mention (e.g. row 198-200; 229-232)

Author Response

The third reviewer (R3) made one question on whether the expression malnutrition in the title includes undernutrition, insufficient nutrient density, and obesity. For sure, we refer to malnutrition in all its form. It includes undernutrition (wasting, stunting, underweight), inadequate vitamins or minerals, overweight and obesity (see page 2).

The reviewer (R3) also asked whether we could specify how many articles were from which database. We would like to emphasize that all articles included in this manuscript were from open access journals. We search for articles in Web of Science, PubMed and Google Scholar, and we found 101 articles; of these, 33 were from Web of Science, 46 from PubMed and 22 were from Google Scholar. After removal of duplicates, we were left with 71 articles as mentioned in the manuscript (see page 3). Altogether, 14 articles were included in the manuscript; of these, 4 were from Web of Science [17, 31, 33, 35], 6 from PubMed [18, 19, 29, 30, 32, 38] and 4 from Google Scholar [28, 34, 36, 37].

The reviewer (R3) further asked how many articles were excluded in additional search. We would like to emphasize that we found 9 articles; of these, 4 were included in the manuscript and 5 did not meet the research criteria for eligibility.

At last, the reviewer (R3) made a comment on the relevance of explaining an unusual result by Kazembe et al [36]. Definitely, it is an unusual result that food-insecure households were found to consume less of unhealthy food. This has now been remarked in the discussion section, and it has already been well explained and supported (see page 14).

This manuscript is a resubmission of an earlier submission. The following is a list of the peer review reports and author responses from that submission.

Round 1

Reviewer 1 Report

“Food insecurity and health outcomes in southern Africa: a systematic descriptive review” identifies empirical studies that related FI to health outcomes in southern Africa. The review is well written but some improvements could be made:

-in introduction, state of the art is missing, the authors can underline the current state of research on this topic presented in the literature

-also, I suggest a graph or/and a statistical model for the prevalence of FI in different regions.  

Reviewer 2 Report

The authors have engaged in an attempt to review health outcomes from food insecurity and submitted it to the biosafety section if the international journal of environmental research and public health. 

The review does not deal with biosafety and suffers from two major weaknesses:

1) Scoping: the approach to food insecurity neglects its more important health outcomes, including in relation to acute severe malnutrition and stunting in children. In addition, the authors focus on mental health and non-communicable diseases, as well as HIV acquisition risk. It is not reasonnable to deal with such specific issues in a review, which ends up with 14 eligible papers (from 101 screened).

2) Content: the authors seem to understand the underlying causes of malnutrition, including malaria, diarrhea and acute respiratory diseases and do not mention it. Same applies to the food safety component of food (in)security, which could have been a focus of this review. Nonetheless, food safety was not presented as such as a topic, nor were health outcomes from eating unsafe food included, whereas food safety is by definition an integral part of food insecurity.

My suggestion to them is narrow down the scope of their review to limit it to the impact of food insecurity on mental health and HIV acquisition in sub-Saharan Africa.

Reviewer 3 Report

The manuscript by Militao, E. et. al. entitled “Food Insecurity and Health Outcomes in Southern Africa: A Systematic Descriptive Review” is a review study with an aim to determine the impact food insecurity (FI) on health outcomes among people living in southern Africa region.

The review included 28 papers/studies.

Some technical comments:

English proofreading is suggested and repetitive sentences should be removed.

One question that need explanation, why Mozambique was chosen as an important area of interest? How many areas of southern Africa region are/aren’t covered by FI surveys?

Row 18 – is Fi major or one of the main?

Row 47/56 – p.4/ p.51 - report that you list in references has only 4 pages

Row 93 – already asked question why Mozambique

Row 146 – “in the last 3 months” – define exact time interval – is this last three months from now or last three months in paper or something else?

Row 120-121 - why did you subsequently added those references? If the answer is that it is because to complete 20 years, why this time interval was not set from the beginning of selection process?

Hypertension, Diabetes, Anxiety, depression and HIV

Part about FI  - one should comment not only results obtained in separate papers but also put results obtained in those papers in the context of the years when they were collected.

In the part where you connect FI with health outcomes try to group similar outcomes together

Row 277-279 – reference missing

Row 298-301 – that is not only situation in Africa but in all middle-income countries around the world.

Row 303 – LMIC (reference)

Row 422-425 – reference missing

You need to give more emphasis on FI on health outcomes in the manuscript because it is set as the main goal of this paper and more attention has been paid to socioeconomic reasons for their FI.